# Considerations for hypothetical carbon dioxide removal via alkalinity addition in the Amazon River watershed

Linquan Mu, Jaime B. Palter, Hongjie Wang

Graduate School of Oceanography, University of Rhode Island, Narragansett, RI, 02882, USA

*Correspondence to*: Linquan Mu (mulinquan@gmail.com)

**Abstract.** The Amazon River plume plays a critical role in shaping the carbonate chemistry over a vast area in the western tropical North Atlantic. We conduct a sensitivity analysis of hypothetical ocean alkalinity enhancement (OAE) via quicklime addition in the Amazon River watershed, examining the response of carbonate chemistry and air-sea carbon dioxide flux to the alkalinity addition. Through a series of sensitivity tests, we show that the detectability of the OAE-induced alkalinity increment depends on

the perturbation strength (or size of the alkalinity addition, $\Delta TA$) and the number of samples: there is a 90% chance to meet a minimum detectability requirement with $\Delta TA > 15$ $\mu$mol kg$^{-1}$ and sample size $> 40$, given background variability of 15–30 $\mu$mol kg$^{-1}$. OAE-induced $p$CO$_2$ reduction at the Amazon plume surface would range between 0–25 $\mu$atm when $\Delta TA = 20$ $\mu$mol kg$^{-1}$, decreasing with increasing salinity (S). Adding 20 $\mu$mol kg$^{-1}$ of alkalinity at the river mouth could elevate the total carbon uptake in the Amazon River plume (15 < S < 35) by at least 0.07–0.1 MtCO$_2$ month$^{-1}$, and a major portion of the uptake would occur in

the saltiest region (S > 32) due to its large size, comprising approximately 80% of the S > 15 plume area. However, the lowest salinity region (S < 15) has a greater drop in surface ocean partial pressure of CO$_2$ ($p$CO$_2$$^{sw}$) due to its low buffer capacity, potentially allowing for observational detectability of $p$CO$_2$$^{sw}$ reduction in this region. Reduced outgassing in this part of the plume, while more uncertain, may also be important for total additional CO$_2$ uptake. Such sensitivity tests are useful in designing minimalistic field trials and setting achievable goals for monitoring, reporting, and verification purposes.

## 1 Introduction

To meet the Paris Agreement goal of limiting global temperature change to well below 2°C (UNFCCC, 2015), reducing greenhouse gas emissions is urgently needed, but insufficient on its own. Modeling results from the IPCC (2022) estimated a total remaining carbon budget of less than 500 GtCO$_2$ for a >50% probability of staying below 1.5°C warming by 2100. Staying within this small remaining budget is a formidable challenge, as it requires current emission rates, which totaled ~36 GtCO$_2$ yr$^{-1}$ in 2021

(Friedlingstein et al., 2021), to rapidly approach zero. Accordingly, even very optimistic emission reduction scenarios assume carbon dioxide removal (CDR) will be needed to remove 10–20 GtCO$_2$ yr$^{-1}$ from the atmosphere by the end of the century (NASEM, 2019; 2021). With the ocean covering ~70% of the Earth's surface and providing the largest sink for anthropogenic CO$_2$ emissions to date, there is growing interest in CDR solutions in the marine environment.

Several ocean-based CDR approaches have been suggested over the past decades to reduce CO$_2$ in the atmosphere (NASEM, 2021).

Enhanced Weathering (EW) and Ocean Alkalinity Enhancement (OAE) are related techniques with the primary goal to accelerate the weathering process that would naturally remove atmospheric CO$_2$ at a very slow rate (10,000 to 100,000 years; Gonzalez and Ilyina, 2016). EW techniques involve pulverization of weatherable rocks and their application on land, while OAE applies those materials to increase alkalinity at the ocean surface (Kheshgi, 1995; Renforth and Henderson, 2017). When mineral additions are made near the edge of a coastal watershed, the weathering process may happen on land and/or in the ocean, and the distinction

between the two techniques blurs.

EW and OAE can have other co-benefits besides reducing $CO_2$ and mitigating global warming. Several early EW experiments via regional-scale liming focused on mitigating acidification in inland lakes and ambient streams. These studies found that the surface water pH could be elevated to desired levels and persist for years without causing deleterious effects to the environment (e.g., Wright, 1985; Porcella, 1989; Driscoll et al., 1996). EW deployments in agricultural settings have also been proven effective in reducing soil acidity, preventing soil erosion, and enhancing crop yields (Caires et al., 2006; Köhler et al., 2010; Kantzas et al., 2022). Similarly, OAE can increase the pH of seawater, alleviating ocean acidification, which is a major stressor for the marine ecosystem (e.g., Doney et al., 2009). Both EW and OAE approaches are still in early phases of conceptualization and most OAE studies so far have focused on numerical simulations or efforts within laboratories (Köhler et al., 2010; Gonzalez and Ilyina, 2016; Moras et al., 2022; Wang et al., 2022). Understanding the feasibility, effectiveness, and ecological risks of OAE is required before any large-scale efforts should be implemented (NASEM, 2021). Though harmful ecosystem effects associated with highly elevated alkalinity cannot be ruled out (Bach et al., 2019), this risk must be weighed against a counterfactual in which carbon dioxide remains in the atmosphere.

Large river-dominated tropical oceans are potential test ground for OAE, for several reasons. First, mixing and subduction of surface waters into the ocean interior is minimized in tropical oceans relative to higher latitudes (Gonzalez and Ilyina, 2016; Lenton et al., 2018). Second, large rivers form surface plumes that extend thousands of kilometers offshore (Lentz and Limeburner, 1995; Coles et al., 2013). In combination, these two factors mean that added alkalinity would have a long time to absorb $CO_2$ at the surface ocean and impact a vast area along the plume path. During this time, the atmospheric $CO_2$ is continuously sequestered by the ocean until a new air-sea $CO_2$ equilibrium is reached. The plumes also have a heightened potential for observational tracking using surface salinity, which can be estimated from satellite observations. Finally, the carbonate-poor river waters (relative to ocean water) may help suppress secondary chemical precipitation of the added alkalinity, a risk that can reduce the efficiency of OAE (Bach et al., 2019; Hartmann et al., 2022). Overall, the transport of alkalinity in rivers to the ocean from EW deployments is seen as a CDR technique with the potential to scale to the gigaton level (Zhang et al., 2022).

Therefore, we examine the Amazon River-ocean continuum for its potential as a site of OAE. As the world's largest river by volume, the Amazon River represents ~20% of the global riverine discharge into the oceans (Salati and Vose, 1984). Its massive outflow (an average of ~0.2 Sverdrup; Figure S1) creates a thin surface layer of low-salinity (Figure 1) and low-carbonate plume, extending up to $1.5 \times 10^6$ km$^2$ at the ocean's surface (Molleri et al., 2010). As a result, the Amazon River plume has profound influence on the carbonate dynamics and atmospheric $CO_2$ sequestration throughout the western tropical North Atlantic Ocean (Ternon et al., 2000; Cooley et al., 2007; Lefèvre et al., 2010; Ibánez et al., 2015; Mu et al., 2021; Olivier et al., 2022; Monteiro et al., 2022).

In this study of hypothetical alkalinity addition at the Amazon River mouth, we investigate the expected changes in air-sea $CO_2$ flux at the offshore Amazon plume waters and examine what the perturbation size and sampling density would be needed to measure the alkalinity change and verify a resultant anomalous $CO_2$ flux. We make use of the knowledge from previous field studies of the carbonate chemistry in the Amazon River plume, to address two main goals: 1) Analyze the potential for detectability of OAE-induced alkalinity change relative to measurement precision, background variability, and sample size; and 2) Estimate changes in ocean $p$CO$_2$ and air-sea $CO_2$ flux in the Amazon River plume due to hypothetical alkalinity addition at river mouth. Our effort aims to outline how one might consider the measurement, reporting and verification (MRV) needed in the context of known background variability. We argue that this kind of sensitivity analysis is a first step that could lead to more realistic numerical simulations if the system does not fail basic tests of feasibility.

## 2 Method

### 2.1 Study Site and mixing model

We use a river-ocean conservative mixing model (Cooley and Yager, 2006) informed by direct observations collected as parts of the ANACONDAS and ROCA projects (Mu et al., 2021; Mu et al., in revision) to establish expectations of how alkalinity perturbations would influence the carbonate system in the Amazon plume (the gray and green pathways on the methods schematic shown in Figure 2). The mixing model assumes no sources or sinks of the carbonate species modeled, and the deviations from this assumption are discussed in Section 2.2. In this model, dissolved inorganic carbon (DIC) and total alkalinity (TA) are treated as conservative tracers and used to describe $p\mathrm{CO_2}$ variations as a function of salinity and temperature (Figure 1). TA and DIC can be respectively expressed as:

$$\mathrm{TA} = [\mathrm{Na^+}] + 2[\mathrm{Mg^{2+}}] + 2[\mathrm{Ca^{2+}}] + \cdots - [\mathrm{Cl^-}] - 2[\mathrm{SO_4^{2-}}] - [\mathrm{Br^-}] - \cdots \qquad (1)$$

$$\mathrm{DIC} = [\mathrm{HCO_3^-}] + [\mathrm{CO_3^{2-}}] + [\mathrm{CO_2}] \qquad (2)$$

We adopt the explicitly conservative form of TA here (Wolf-Gladrow et al., 2007) so that TA changes due to the addition of our hypothetical calcium-based alkalinity feedstock can be fully tracked by the increase in calcium ion concentration. To establish the baseline condition of the carbonate system in the Amazon River-ocean continuum, we adopted the river endmembers from Mu et al. (in revision; also, Table 1) and ocean endmembers (S = 36, TA = 2,369 $\mu$mol kg$^{-1}$, DIC = 2,025 $\mu$mol kg$^{-1}$) from Mu et al. (2021). The conservative mixing model can be further expressed as:

$$SSS_{mix} = S_r \times f_r + SSS_o \times f_o \qquad (3)$$

$$f_r + f_o = 1 \qquad (4)$$

$SSS_{mix}$ is the salinity for a plume sample; $S_r$ is the salinity of the river endmember ($S_r = 0$); and $SSS_o$ is the surface salinity of the ocean endmember. In Equation (4), $f_r$ and $f_o$ are the proportions of river and ocean in the sample, both of which can be solved when $SSS_{mix}$ is known. The theoretical TA and DIC in the plume, TA$_{mix}$ and DIC$_{mix}$ at any given $SSS_{mix}$ are then calculated from the

95 known endmember properties and proportions via:

$$\mathrm{TA_{mix}} = \mathrm{TA_r} \times f_r + \mathrm{TA_o} \times f_o \qquad (5)$$

$$\mathrm{DIC_{mix}} = \mathrm{DIC_r} \times f_r + \mathrm{DIC_o} \times f_o \qquad (6)$$

Eventually, $p\mathrm{CO_2}^{mix}$ and pH$_{mix}$ at a given salinity and observed temperature are calculated with CO2SYS (Lewis et al., 1998) from inputs of TA$_{mix}$ and DIC$_{mix}$. In this calculation we use the carbonic acid dissociation constants K1 and K2 from Millero (2010) due

to their eligibility across a wide salinity range suitable for this study.

We confined our study periods to two specific months, September 2011 and July 2012, during which measurements were made in both the river and throughout the Amazon River plume. During the river plume/ocean expeditions, surface seawater $p\mathrm{CO_2}$, temperature, salinity, and chlorophyll fluorescence were continuously measured shipboard across the Amazon-influenced regions; An oceanic station was specifically sampled for TA and DIC during each cruise as the ocean carbonate endmember for each

105 corresponding month (Mu et al., 2021). Details in quantifying the river endmembers are described in Ward et al. (2015) and Mu et al. (in revision). Briefly, the Amazon River mouth region receives freshwater runoff from three channels: the north and south channels near Macapá, and the channel of the Tocantins River near Belém (Figure S2). Sampling was conducted at the lower reaches of the three channels near the mouth (or "gateways") in each month. Six TA and DIC samples were collected at three cross-channel locations of each gateway (left bank, center, and right bank) from both surface and 50% depth (5–10 m). For a

concise demonstration on the detectability of the alkalinity addition in the river, we chose to target one gateway (North Macapá) as the perturbation site and consider how many samples would be needed to observe the TA perturbation relative to that gateway's background variability. However, to quantify the potential CDR induced by a TA perturbation just above detectability, we assumed the entire river mouth would be perturbed equally as in North Macapá. Because the outflow through both North and South Macapá gateways comprise > 80% of total discharge (Ward et al., 2015; Mu et al., in revision) and their chemical properties are almost identical seasonally (Mu et al., in revision), North Macapá is representative of the mouth's chemical composition. In an actual OAE field trial, perturbing one gateway will perturb the mouth region by an amount proportional to the percentage discharge there (relative to the combined discharge), and the natural variability in the unperturbed gateways will affect the background variability at the mouth. There are key unknowns in seasonal TA contributions from other gateways to the mouth which prevents us from combining all the gateways. Therefore, the mean and standard deviation of the six TA values at North Macapá, respectively, were then used as the river TA endmember and its natural background variability (Table 1). The monthly $CO_2$ fluxes in the plume for both chosen months are displayed in Table 2.

## 2.2 Deviations from the conservative mixing model

The assumption that TA and DIC act as conservative tracers is not true in the presence of photosynthesis, respiration, bio-calcification, abiotic dissolution/precipitation of calcium carbonate, and/or air-sea gas exchange. In fact, plume $pCO_2$ values measured from an underway system on the research vessels surveying for the ANACONDAS program were systematically lower than predicted from the mixing model (Mu et al., 2021; Figure S3). This offset was assumed to be due to net community production, an assumption supported by the strong correlation between shipboard chlorophyll and the difference between the $pCO_2^{mix}$ calculated from conservative mixing and the measured $pCO_2$ (Mu et al., 2021).

A linear regression between measured SSS ($15 < SSS < 35$) and measured $pCO_2$ showed a strong linear relationship (r = 0.79 for September 2011 and r = 0.97 for July 2012). Therefore, we use SSS to calculate an empirical estimate of $pCO_2$ ($pCO_2^{empirical}$) at every SSS level in each month (orange pathway in the schematic shown as Figure 2). SSS fields in the Amazon plume were derived from remotely sensed diffuse attenuation coefficient at 490 nm ($K_d490$, Figure 1). The measured-SSS vs $pCO_2^{empirical}$ regression is used to map $pCO_2^{empirical}$ at every satellite-derived SSS value in the plume (as in Figure 3a) and to calculate the empirical air-sea $CO_2$ flux, according to the following equation:

$$Flux_{empirical} = k\alpha(pCO_2^{empirical} - pCO_2^{atm}) \qquad (7)$$

Where k is the gas exchange coefficient calculated from gridded reanalysis wind speed and the parameterization of Sweeney et al. (2007), $\alpha$ is the solubility of $CO_2$ in seawater (a function of water temperature and salinity; calculated from equations in Weiss, 1974), and $pCO_2^{atm}$ is the monthly averaged $pCO_2$ in the atmosphere measured at a nearby NOAA monitoring station. Details in calculating the empirical $CO_2$ flux can be found in Mu et al. (2021).

## 2.3 Experimental Design

The core of this sensitivity analysis is to manipulate the concentrations of TA at the river mouth by the addition of a hypothetical alkalinity feedstock. There are several feedstocks that could be considered for this purpose, each with a distinct set of considerations (Hartmann et al., 2013; Renforth and Henderson, 2017). Magnesium or silicate-rich rocks such as olivine and other basalts (Hartmann et al., 2013), carbonate minerals (e.g., Rau, 2011; Renforth et al., 2022), mineral derivatives found in some industrial waste products, or even electrochemical production of alkalinity (Tyka et al., 2022; He and Tyka, 2023) are all candidates (NASEM, 2021). The stoichiometry of the reaction, dissolution rate, risk of secondary precipitation, and contamination with metals are all

specific to the particular alkalinity source chosen. Quicklime (CaO) has been identified as being promising, due to its high solubility in sea water and rapid dissolution rates (Kheshgi, 1995; Bach et al., 2022). Quicklime is made by heating limestone (primarily $CaCO_3$), an abundant rock at the Earth's surface. The heating of each molecule of $CaCO_3$ releases one molecule of quicklime and

150 one molecule of $CO_2$ (Kheshgi, 1995). Therefore, carbon capture and storage of the released $CO_2$ would be necessary for OAE to be effective from the application of quicklime (Paquay and Zeebe, 2013; Renforth et al., 2013; NASEM, 2021; Moras et al., 2022). Hereafter, we assume that the alkalinity feedstock for the hypothetical OAE deployment is quicklime, which simplifies the discussion of stoichiometry, precludes contamination from metals and other industrial contaminants, justifies our assumption of rapid dissolution, and informs our discussion of potential unintended consequences.

Assuming the additional CaO converts 100% to TA according to stoichiometry, the dissolution of CaO consumes $CO_2$ and releases $HCO_3^-$ through:

$$(CaO + H_2O \rightleftharpoons Ca(OH)_2) + 2\,CO_2 \rightleftharpoons Ca^{2+} + 2\,HCO_3^- \qquad (8)$$

For every mole of additional CaO, TA is increased by 2 moles (Equation 1) while DIC remains constant (i.e., consuming 2 moles of $CO_2$ while producing 2 moles of bicarbonate in Equation 8) before any significant air-sea equilibration occurs. This assumption

of a constant DIC during perturbation should hold over the time scale of the river-ocean mixing based on the following evidence: 1) CaO dissolution happens on hour-scales (Moras et al., 2022); 2) The time elapsed between the river outflow at the mouth and its mixing with ocean to relatively high salinity (SSS > 15) is as short as two weeks (c.f., Figure 5 from Coles et al., 2013), while the air-sea $CO_2$ equilibration usually takes several weeks to months in the western tropical North Atlantic (Jones et al., 2014). We assume in this study that there is no secondary precipitation of $CaCO_3$, and that the additional CaO stays at the surface layer until

fully dissolved. Another underlying assumption is that CaO is homogeneously mixed across the river mouth gateway, eliminating potential uncertainty / variability due to patchiness of the TA addition. We will address key unknowns associated with these assumptions in the Discussion.

We explore TA perturbations over the range 1–100 $\mu$mol kg$^{-1}$ (i.e., perturbation between 0.07 and 7 times the natural variability for September 2011 and 0.04 and 4 times the natural variability for July 2012). The mixing model is recalculated with each of the

170 perturbed river endmembers to find the perturbed $TA_{mix}$ and $DIC_{mix}$ across the plume. Then we use CO2SYS to calculate the theoretical $pCO_2^{mix\_OAE}$ at each salinity stamp with the perturbed $TA_{mix}$, $DIC_{mix}$ and observed temperature as inputs to compare to the unperturbed baseline. The difference of the TA-enhanced seawater $pCO_2$ and the baseline $pCO_2$:

$$\Delta_{OAE}pCO_2^{mix} = pCO_2^{mix\_OAE} - pCO_2^{mix\_base} \qquad (9)$$

In each perturbation scenario, the added alkalinity lowers the $pCO_2^{mix\_OAE}$ below the unperturbed mixing curve by the amount of

175 $\Delta_{OAE}pCO_2^{mix}$. To finally arrive at the surface ocean $pCO_2$ needed to calculate the air-sea fluxes according to Equation 7, we add the $\Delta_{OAE}pCO_2^{mix}$ to $pCO_2^{empirical}$:

$$pCO_2^{OAE} = \Delta_{OAE}pCO_2^{mix} + pCO_2^{empirical} \qquad (10)$$

This approach (schematized in Figure 2) implicitly assumes that the biological processes that lower the measured $pCO_2^{empirical}$ below the theoretical $pCO_2^{mix}$ would be unchanged from the deployment of additional alkalinity, an assumption that would demand

careful testing in the field.

**2.4 Sensitivity evaluation for detectability of hypothetical OAE deployments**

To assess the TA detectability, we evaluate perturbations in one river gateway (North Macapá). We use a simple bootstrapping technique (or Monte Carlo simulation) to assess the minimum TA perturbation that would be detectable with a given number of measurements, taking into consideration the background variability during a given season and reasonable measurement precision.

We randomly generate 1,000 baseline TA values under a normal distribution using the mean and standard deviation of the six measured TA values in September 2011 or July 2012. We then generate 1,000 perturbed TA values for each perturbation scenario under a normal distribution, where the mean is the addition of the baseline mean and the perturbation strength, with the same standard deviation as the baseline. The range for the tested perturbation sizes is 1–20 $\mu$mol kg$^{-1}$ of TA. Within the 1,000 TA pools from both the baseline and a perturbation scenario, we randomly select 1–100 values (i.e., varying sample size) from both pools, perform the Student's t-test between the selected sets, and calculate the p-value to determine if the perturbed TA set would be seen as significantly different from baseline. The t-test is repeated 200 times for each perturbation size and sample size, generating 200 p-values. The mean of those p-values are calculated for each perturbation scenario at each sample size (Figure 4). A lower p-value suggests higher likelihood that the perturbed TA values would be considered significantly different from the baseline, and therefore "detectable". We use a threshold of $p < 0.1$ to indicate statistical significance.

**3 Results**

**3.1 Baseline carbon chemistry in the plume**

The surface $pCO_2^{empirical}$ in the Amazon River plume derived from remotely sensed SSS generally ranged from 210–400 $\mu$atm in September 2011 (Figure 3a), with lowest values in low-salinity regions near French Guiana coast and increasing as the river mixes with ocean water (Figure 1a). Figure 3a masks out regions with surface salinity below 15 and beyond 35 psu, as these waters are out of the range of a sufficiently robust algorithm between $K_d490$ and SSS (Mu et al., 2021).

The distribution of plume surface $pCO_2^{sw}$ is primarily shaped by two processes: the Amazon River-ocean mixing and biological $CO_2$ consumption/release in the plume waters (Mu et al., 2021). For example, zero-salinity river water near the mouth is supersaturated with respect to atmospheric $CO_2$ in July 2012, where $pCO_2 > 1,000$ $\mu$atm is observed (Mu et al., 2021; Mu et al., in revision) due to low carbonate buffer capacity and high microbial respiration. As the river waters mix with the ocean, the sharp increase in buffer capacity and shift towards a net autotrophic state lower the surface $pCO_2$ towards a minimum (below $pCO_2^{atm}$ in July 2012; Mu et al., 2021) before rising towards the open ocean levels closer to equilibrium with the atmosphere (i.e., $pCO_2$ ~400 $\mu$atm) at higher salinity (Figure S3). The nitrogen-fixing diatom-diazotroph assemblages and other phytoplankton that are active in the Amazon plume waters (Goes et al., 2014) further enhance the $CO_2$ undersaturation in mid- to mid-high salinity portions of the plume (i.e., $15 < SSS < 33$) on top of the undersaturated $CO_2$ state caused by conservative mixing (Mu et al., 2021).

**3.2 Detectability of TA perturbations**

We propose that the minimum requirement for MRV in a watershed OAE experiment is that the perturbed TA in the river can be detected above background variability. To illustrate the challenge of detectability, we consider the background TA variability in the North Macapá gateway in September 2011, with the assumption that the variability measured at that time (standard deviation 14.5 $\mu$mol kg$^{-1}$) is representative of that season and gateway. It is intuitive that a large enough perturbation (commensurate with the standard deviation) is detectable with a small number of samples (Figure 4), and a small perturbation (a factor of 3 smaller than the standard deviation or smaller) cannot be detected against the background variability regardless of the number of samples. This

exercise reveals the challenge of balancing the effort needed to detect the perturbation against the obstacle and risk of increasing the TA by a large amount.

As expected and illustrated in Figure 4, higher sample sizes and TA perturbations lead to greater detectability of the alkalinity enhancement. For practical purposes, neither sample size nor perturbation strength can be increased infinitely, so OAE experiments would likely seek a balance between the two. For example, in September 2011 (Figure 4a, c), a TA perturbation of ~7–10 $\mu$mol kg$^{-1}$ at a background standard deviation of 14.5 $\mu$mol kg$^{-1}$ would have been readily detectable with 40 samples. For July 2012 (Figure 4b, d), when background variability was higher (25.7 $\mu$mol kg$^{-1}$), 40 samples would detect perturbations only if they were to exceed +20 $\mu$mol kg$^{-1}$ of TA. In an actual field OAE trial, each season (and gateway) may have different background variability that would require measurement and characterization in advance of any perturbation to have a clear strategy for sampling.

### 3.3 Impact on $p$CO$_2$ and air-sea CO$_2$ flux

Figure 3b shows the $p$CO$_2$$^{mix}$ decrease due to a hypothetical TA addition in the Amazon watershed, which is highest in the low-salinity waters near the river mouth and attenuates at higher salinities due to mixing with the unperturbed ocean water. Figure 5 provides a quantification of the air-sea CO$_2$ flux at each salinity level for different TA perturbation strengths (20, 50 and 100 $\mu$mol kg$^{-1}$). While the air-sea CO$_2$ flux per unit area (or flux density) is greatest at low salinities (Figure 5a, b), the large area occupied by the diluted plume (salinities greater than 32; Figure S4) means that more than half of the total integrated air-sea CO$_2$ exchange – as well as the perturbation to this flux – would occur in this salty part of the plume.

Air-sea CO$_2$ exchange at SSS = 34–35 for July 2012 is only slightly greater than 0 (Figure 5b), but due to the large area of plume at near-oceanic salinity level, the total plume CO$_2$ uptake (i.e., negative flux) in lower S regions is entirely offset by the CO$_2$ outgassing at 34 < SSS < 35. Regardless of whether the baseline Amazon plume is a carbon sink or source, the change in air-sea CO$_2$ exchange due to OAE shifts toward more carbon storage by the ocean and increases linearly with the size of TA perturbation (Table 2).

## 4 Discussion

### 4.1 Unaccounted for plume region with great CDR potential

While we explored the OAE-induced additionality of the carbon uptake (or net CDR) at different salinities in the Amazon River plume (Figure 5), it is important to note that the cumulative additional uptake in Figure 5b excludes the freshest part of the plume (SSS < 15). Due to high organic carbon remineralization in shallow waters (Mu et al., 2021), the quasi-linear SSS versus $p$CO$_2$ empirical relationship collapses at SSS < 15 and prevents the establishment of empirical $p$CO$_2$ in this region. Therefore, we excluded SSS < 15 in our analyses (e.g., in Figure 3, coastal areas < 50 m deep near the mouth are masked) while acknowledging that because of the strong CO$_2$ outgassing in this region, the baseline air-sea CO$_2$ flux in the entire plume (0 < SSS < 35) will be shifted towards more CO$_2$ release, should the SSS < 15 water be included. However, once TA is added, surface $p$CO$_2$ will decline in the entire plume regardless of salinity, causing greater net CO$_2$ drawdown across the plume. In other words, the OAE-induced CO$_2$ uptake increase calculated for 15 < S < 35 plume water could substantially underestimate the true additionality when SSS < 15 is not considered. The area of the S < 15 water is small compared to the 15 < SSS < 35 plume (i.e., < 15% of the SSS > 15 plume area), but its lower buffer capacity also means $p$CO$_2$ is much more sensitive to TA addition, and therefore could still contribute a considerable amount to CDR.

A basic scaling would suggest that 10% of the plume area with an OAE-induced surface $pCO_2$ decrease of 30 $\mu$atm would have as large an impact on area-integrated ocean uptake as the $15 < SSS < 35$ portion of the plume with a 3 $\mu$atm decrease. Therefore, including the freshest part of the plume and calculating its reduction in outgassing could easily double our estimate of additional $CO_2$ uptake for the entire plume. Lacking both knowledge of the spatial distribution of salinity and a robust estimate for $pCO_2$ in this low salinity region prevents us from quantifying its CDR potential in a rigorously manner.

However, we can still conduct a back of the envelope calculation to estimate the distribution of CDR between $SSS < 15$ and $15 < SSS < 35$ regions of the plume. With an addition of 20 $\mu$mol kg$^{-1}$ TA and an averaged Amazon freshwater discharge of 0.18 Sverdrup (Figure S1), we can calculate the total transport of this TA addition of $9.3 \times 10^9$ mol month$^{-1}$, given sufficient time for air-sea re-equilibration and the subsequent DIC increase due to OAE. If we assume the maximum DIC increase is 0.8 times the $\Delta$TA (Wang et al., 2022), the maximum CDR resulting from TA addition would be $0.8 \times 9.3 \times 10^9$ mol month$^{-1}$ = $7.5 \times 10^9$ mol month$^{-1}$, or 0.3 MtCO$_2$ month$^{-1}$. Given the estimated enhanced $CO_2$ flux in the $S > 15$ region is approximately 0.07~0.1 MtCO$_2$ month$^{-1}$ (Table 2), ~0.2 Mt of $CO_2$ uptake per month would be expected to take place in the low salinity region ($SSS < 15$), which is consistent with the estimation based on direct air-sea $CO_2$ flux change. This calculation highlights the potentially important role of the freshest part of the plume in the net CDR by the entire plume, which could be disproportionately large relative to its small size.

## 4.2 Exploring the minimum TA detectability

Here we propose that the minimum MRV requirement for an OAE experiment in a river be that the TA increase can be detected in the fresh river endmember, and we showed what size perturbation and sampling density would meet that minimum for the Amazon River. We found that the minimum detectable TA addition at the Amazon River mouth ranges between 10–20 $\mu$mol kg$^{-1}$ with a reasonable sample size of 40 during field campaigns before and during the perturbation (Figure 4c, d). We focus on detectability results for TA, since that is the variable directly targeted by any OAE approach. However, we present similar detectability analyses for the increase in pH (Figure S5) and decrease in $pCO_2$ in the river mouth (Figure S6) in the Supplementary Materials, which shows that the perturbation to these variables from a 10–20 $\mu$mol kg$^{-1}$ is also readily detectable with a reasonable number of samples. With commercially available autonomous sensors for pH and $pCO_2$ (Kapsenberg et al., 2017; Sabine et al., 2020), these might be attractive proxies for the added TA.

One may further hope that MRV efforts for OAE deployed at the river mouth would document reductions in plume $pCO_2$. We rely on the observed $pCO_2$, a mixing model, and satellite-derived SSS to deduce where one might expect $pCO_2$ reductions to be measurable relative to background concentrations. Sensors for measuring $pCO_2^{sw}$ aboard ships and uncrewed surface vehicles like Saildrone and Waveglider have a target accuracy of 2 $\mu$atm (Sabine et al., 2020). Figure 3 shows that, if Amazon River water TA were continuously increased by 20 $\mu$mol kg$^{-1}$, a broad region of the plume would see $pCO_2^{sw}$ decreases measurable with current sensor technology, if they could sustain regular sampling over the $SSS < 30$ region before, during, and after the OAE deployment.

However, like with the detectability of the TA in the river mouth, the $pCO_2$ perturbation must be detectable given background variability, and does not depend only on sensor accuracy. Here, a measure of that background variability is the root-mean-square error (RMSE) between the $pCO_2^{empirical}$, calculated from the linear regression of the *in situ* observations of $pCO_2$ and SSS, and the actual observations of $pCO_2$ ($pCO_2^{measured}$, Figure 2). For September 2011, the RMS error of this regression is ~22 $\mu$atm for $15 < SSS < 35$ (Figure S7). Thus, we would expect that a perturbation approximately this size would be detectable with sufficient samples at the river mouth (Figure S6). Figure 3 and S7 suggest that a region of the plume near the river mouth approximately $15 < SSS < 20$ could meet this criterion due to 1) lower salinity and higher perturbed TA and 2) relatively lower RMS error in

$pCO_2^{empirical}$ and thus low background $pCO_2$ variability. This speculation is nicely verified in a minimum OAE detectability test using $pCO_2$ in the plume waters as the proxy (Figure S8), where the reduction in $pCO_2$ at $\Delta TA = +10$ $\mu$mol kg$^{-1}$ is readily detectable with 10 samples for $15 < SSS < 20$ waters (Figure S8c), contrasting sharply with the poor detectability at higher salinity ($30 < SSS < 35$; Figure S8d).

It is encouraging that SSS can be estimated from satellite measurements, allowing for field campaigns – including uncrewed
vehicles piloted from operators onshore – to find and sample in the plume where an OAE-induced $pCO_2$ reduction may be detected. Nevertheless, Figure 5 shows that nearly half the additional ocean $CO_2$ uptake due to the OAE experiment could happen in the part of the plume with $30 < SSS < 35$, where detecting the perturbation in TA and plume surface $pCO_2$ is impossible with the accuracy of existing sensors or the laboratory analysis of bottle measurements.

Finally, we consider the amount of limestone ($CaCO_3$) required to produce pulverized quicklime as the alkalinity source material
that sustains a set TA perturbation size (Table 2). The limestone equivalents needed to create a detectable perturbation of $+20$ $\mu$mol kg$^{-1}$ of TA is 0.4 Mt per month using river discharge numbers from September 2011, which is approximately equal to 0.04% of US industrial limestone production in the year 2021 (USGS, https://pubs.usgs.gov/periodicals/mcs2021/mcs2021-stone-crushed.pdf). Using river discharge for July 2012, the requirement grows to 0.7 Mt limestone per month. In terms of OAE-induced additional $CO_2$ uptake, every 0.1 MtCO$_2$ month$^{-1}$ represents more than 30 times greater CDR than the $CO_2$ captured by one ORCA
Direct Air Capture plant in a full year (https://en.wikipedia.org/wiki/Orca_(carbon_capture_plant)), yet amounts to less than what Brazil emitted on an average 2 hour basis in 2021 (https://www.icos-cp.eu/science-and-impact/global-carbon-budget/2021). It is worth noting that concerns have been raised over the feasibility of CaO liming considering the energy need and $CO_2$ release associated with the production of this alkalinity source (e.g., Paquay and Zeebe, 2013; Renforth et al., 2013; Voosen, 2022), but limestone calcination coupled with heat recovery and carbon capture & storage technologies is found to significantly reduce energy
costs and $CO_2$ emission during CaO production, potentially making this liming approach more sustainable (Foteinis et al., 2022).

### 4.3 Constraints on OAE deployments

So far, we have focused most of our efforts on describing the minimum detectable OAE perturbation. It is also important to consider what might constrain the upper limit of TA addition (i.e., through liming), such as ecosystem disturbances and secondary precipitation of calcium carbonate. Ideally, OAE would not put the ambient marine ecosystems at risk. The minimum detectable
perturbation described above is close to the background TA variability, and so has a greater likelihood for small impact on the ecosystem relative to larger deployments. Heavy metals in some source alkaline minerals are problematic (NASEM, 2021), a consideration avoided with the use of 100% pure CaO compound as the source of TA addition. The highest perturbation we explored was $+100$ $\mu$mol kg$^{-1}$ of TA added to the river endmember, which would lead to initial pH increase of 2.0 at the river mouth, a $pCO_2^{sw}$ decrease of up to 100 $\mu$atm (data not shown) in the Amazon plume, and a total enhanced atmospheric carbon
uptake of ~0.35 MtCO$_2$ month$^{-1}$ across the plume ($15 < SSS < 35$, with considerable additional contribution likely in the freshest plume waters) during September 2011 (Table 2 and Figure 5c). This level of initial pH increase at the TA addition site is likely concerning for local marine communities, although the effect would be dispersed by the river and diminish with increasing distance from the river due to mixing.

Another constraint on the upper limit of TA addition through ocean liming is secondary $CaCO_3$ precipitation and the resulting
reduction of atmospheric $CO_2$ uptake. Moras et al. (2022) suggested that runaway $CaCO_3$ during OAE may be avoidable if the saturation state of aragonite ($\Omega_A$) is kept below 5 for natural seawater. Assuming such a $\Omega_A$ threshold for barely avoiding $CaCO_3$ precipitation is also applicable in fresher Amazon plume and river water, we have calculated the corresponding maximum TA

perturbation allowed to keep $\Omega_A$ below this threshold according to the method in Mucci (1983), with the results shown in Table 3. Because the $\Omega_A$ increases with salinity, TA perturbation that likely avoids secondary $CaCO_3$ precipitation becomes substantially smaller close to SSS = 35 (i.e., ~100 $\mu$mol kg$^{-1}$, compared to ~260 $\mu$mol kg$^{-1}$ at S = 0). Adding 20 $\mu$mol kg$^{-1}$ of TA in September 2011 would only increase $\Omega_A$ by a maximum of 0.3 (Table 3), and $\Omega_A$ stays below 5 everywhere in the plume for TA additions of 100 $\mu$mol kg$^{-1}$ or smaller. An important caveat on assuming a $\Omega_A$ threshold of 5 is that the Amazon River at the mouth contains heavy loads of mineral particles and will likely facilitate the formation of heterogeneous nucleation and precipitation for $CaCO_3$ (Renforth and Henderson, 2017; Moras et al., 2022), thereby lowering the maximum TA addition allowed. Further study is needed to ensure $\Omega_A$ is always kept below proper precipitation threshold in all regions of the river plume.

Adding TA to the land surrounding the Amazon watershed could have possible advantages over spreading across the open ocean, involving co-benefits to the terrestrial ecosystems and agriculture, such as increased crop yield in agricultural watersheds (Caires et al., 2006; Hartmann et al., 2013; Beerling et al., 2020; Kelland et al., 2020), reduced soil run-off (Taylor et al., 2016), and reduction of $N_2O$ production (Beerling et al., 2018). Introducing the TA on land might also slow its dissolution in the river and allow its reaction with $CO_2$ to form bicarbonate in pore water , helping reduce the risks associated with large perturbations to pH and TA, like secondary precipitation or phytoplankton community perturbations. Basaltic rock with slower dissolution rates, but more favorable stoichiometry from CDR than carbonates (Beerling et al., 2018), might also provide an alternative TA feedstock for land-based deployment. However, deployment on land would likely increase the difficulty to quantify the effect of TA addition for MRV purposes.

## 4.4 Additional challenges

One of the underlying assumptions for this analysis is that CaO reacting with $CO_2$ takes place in a time frame faster than air-sea re-equilibration to occur, and the layer of plume water dwells at the surface for a relatively long period of time. If the TA-enhanced surface plume is not exposed to the atmosphere long enough to allow for full air–sea $CO_2$ equilibration (e.g., due to water mass subduction), $CO_2$ removal efficiency will decrease in these regions (Jones et al., 2014). A logical next step to characterize the spatial footprint of the hypothetical experiment and the total time scale for equilibration would be to simulate the time-evolving patch of the alkalinity anomaly, the corresponding $pCO_2$ perturbation, and the total excess DIC in an ocean general circulation model with the OAE experiment relative to a control simulation.

We argue that the TA-enhanced waters do not need to remain continuously within the surface layer to induce ocean CDR. The change in the air-sea flux can be realized whenever the perturbed waters influence the surface, even if that is far from the plume or years after the deployment. Though the implied dilution of the signal or delay in causing additionality of the CDR may not change the net impact of the OAE experiment if integrated over a sufficiently long time period, it would make MRV of the total impact essentially impossible through observational methods alone.

Another assumption in this work was that the photosynthesis, which is known to drive $pCO_2$ beneath the conservative mixing curve (Mu et al., 2021), would be unchanged by the alkalinity addition. This assumption is premised on the argument that perturbations barely detectable above background variability are unlikely to strongly change the ambient ecosystem, an argument supported by a number of recent studies. For example, results from a mesocosm experiment with Tasmanian coastal waters show statistically significant, though relatively moderate, changes to phytoplankton community structure and function at high TA perturbation values of +495 $\mu$mol kg$^{-1}$, which is 25 times higher than the minimum detectable perturbation explored here (Ferderer et al., 2022). Another microcosm incubation study reveals the alkalinity enhancement inducing minor but largely negligible short-term effects

on primary productivity and no measurable changes in biological $CaCO_3$ precipitation, even at a TA perturbation level of +2000 $\mu$mol kg$^{-1}$ (Subhas et al., 2022). Regardless, ecosystem impacts would require careful monitoring in any field OAE experiment.

If the alkalinity source for OAE is quicklime, managing the heat released through its exothermic reaction during dissolution in water at a rate of 64 kJ mol$^{-1}$ CaO is another important consideration. Taking the highest perturbation scenario explored in Table 2 that increases TA in the Amazon watershed by 100 $\mu$mol kg$^{-1}$, we estimate 310 MW total heat released during the duration of the
370 dissolution. In order for the heat released from the CaO reaction to be lower than 10% of the daily solar radiation at the equator ~400 W m$^{-2}$ (a level that would presumably be swamped by natural variability in the river's heat budget), the TA injection would have to occur over an area of river larger than about 8 km$^2$, e.g., 2 km of river width and 4 km along its length. Therefore, it is possible to minimize the environmental risk due to heat released during quicklime dissolution by spreading it over a large enough area at the river mouth.

## 375 5 Conclusions

We conduct a sensitivity analysis of alkalinity enhancement in the Amazon River watershed, evaluating the detectability of added TA, and predicting its influence on the atmospheric $CO_2$ uptake in the offshore plume. Adding 20 $\mu$mol kg$^{-1}$ of TA in a month at the Amazon watershed could increase the $CO_2$ uptake by the river plume by at least 0.07–0.1 MtCO$_2$ month$^{-1}$. The full CDR might be as much as three times as high as this estimate due to reduced outgassing in the freshest part of the plume where we lack a
380 robust empirical $pCO_2{}^{sw}$-salinity relationship and therefore cannot robustly quantify the anomalous air-sea exchange in this region. We found a TA perturbation of +10–20 $\mu$mol kg$^{-1}$ in the river is readily detectable with at least 40 samples, given background variability of 15–30 $\mu$mol kg$^{-1}$. By adding a TA perturbation barely detectable relative to the background TA variability, the likelihood of substantial environmental disturbance at the injection site would be minimized. However, observing the corresponding $pCO_2$ perturbation in the plume presents a greater observational challenge. Even with the highly sensitive $pCO_2$
sensors available today (e.g., Sabine et al., 2020), detectability of the $pCO_2$ anomaly is plausible only in the fresher part of the plume given dilution of the signal at higher salinities and large background variability. Quantifying the total additional $CO_2$ uptake from this type of OAE cannot rely on observations alone; idealized conceptual models like the one presented here and more sophisticated circulation-biogeochemical ocean models will always be required to understand the total CDR because the additional $CO_2$ uptake is likely to occur over long time periods and at very diluted perturbation levels. It is also clear that the basic scientific
feasibility we aimed to assess here does not reveal the full feasibility of such interventions: ocean-based CDR research and deployment require care and attention to ethical, political and governance issues.

### Data availability

Data for the air–sea $CO_2$ flux calculations in the Amazon River plume are from Mu et al. (2021). River mouth data are reported in a manuscript currently under revision in *Global Biogeochemical Cycles* that will be resubmitted in May 2023. All codes are
395 available at https://drive.google.com/drive/folders/1YYjcT5ZYvyHoJaKmRl3Fd8aagMhtRwCA?usp=share_link.

### Author Contribution

JBP and LM conceptualized the framework of this experiment. LM, JBP, and HW designed the methodology. LM performed the analytical calculations and numerical simulations. LM wrote the manuscript with support from JBP and HW. JBP supervised the project.

## Competing Interests

The authors declare that they have no conflict of interest.

## Acknowledgement

We would like to thank Patricia Yager for leading the ANACONDAS and ROCA projects that lay the foundation of this manuscript. LM thanks Jessica Cross, Brendan Carter, Charly Moras, Sijia Dong, and Bo Yang for contributing to discussions, and Sarah Nickford for assistance with figure production.

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

 **Tables**

Table 1. Baseline total alkalinity ($\mu$mol kg$^{-1}$) determined at the N. Macapa gateway at the Amazon River mouth during September 2011 and July 2012 (Mu et al., in revision). The means represent the river's alkalinity endmember in that month.

| Sample No. | Total Alkalinity ($\mu$mol kg$^{-1}$) | |
|---|---|---|
| | **September 2011** | **July 2012** |
| 1 | 279.8 | 362.3 |
| 2 | 286.8 | 322.2 |
| 3 | 310.8 | 301.2 |
| 4 | 296.3 | 327.7 |
| 5 | 307.5 | 296.9 |
| 6 | 275.5 | 295.8 |
| **Mean (std)** | **292.8 (14.5)** | **317.7 (25.7)** |

Table 2. Summary table for air-sea $CO_2$ flux in the Amazon River plume ($15 < SSS < 35$) and the mass of $CaCO_3$ mineral required for each TA perturbation size ($\Delta$TA) in each month. Positive fluxes indicate $CO_2$ outgassing and negative fluxes (–) indicate ocean $CO_2$ uptake. Note that the post-perturbation $CO_2$ flux and $CaCO_3$ demand increase linearly with increase in $\Delta$TA.

| | | September 2011 | | | July 2012 | | |
|---|---|---|---|---|---|---|---|
| | | CO$_2$ flux | | CaCO$_3$ | CO$_2$ flux | | CaCO$_3$ |
| | | mmol m$^{-2}$ d$^{-1}$ | MtCO$_2$ mo$^{-1}$ | Mt mo$^{-1}$ | mmol m$^{-2}$ d$^{-1}$ | MtCO$_2$ mo$^{-1}$ | Mt mo$^{-1}$ |
| $\Delta$TA ($\mu$mol kg$^{-1}$) | **0** | -0.43 | -0.47 | 0 | 0.26 | 0.24 | 0 |
| | **10** | -0.46 | -0.50 | 0.2 | 0.20 | 0.19 | 0.3 |
| | **20** | -0.49 | -0.54 | 0.4 | 0.15 | 0.14 | 0.7 |
| | **50** | -0.60 | -0.65 | 1.1 | -0.02 | -0.01 | 1.7 |
| | **100** | -0.76 | -0.83 | 2.2 | -0.27 | -0.26 | 3.5 |

Table 3. Theoretical aragonite saturation state ($\Omega_A$) at the surface of the Amazon River-ocean continuum, calculated from the river-ocean mixing model (see Section 2.1 and Figure 2) for different TA perturbation strengths ($\Delta$TA) at various salinities in September 2011 using constant water temperature of 29 °C, according to the equation $\Omega_A = [Ca^{2+}][CO_3^{2-}] / K_{sp}^{A(*)}$. $[Ca^{2+}]$ is calculated by adding the additional $[Ca^{2+}]$ due to TA perturbation (i.e., $\Delta$TA/2) to the unperturbed $[Ca^{2+}]$; the latter term is comprised of relative contributions from both the river and ocean $[Ca^{2+}]$ endmembers at a specific salinity. $[Ca^{2+}]$ of 0.15 mmol kg$^{-1}$ is used as the river endmember approximating results from Drake et al. (2021), and the global ocean average $[Ca^{2+}]$ of 10.28 mmol kg$^{-1}$ is used as the ocean endmember. $[CO_3^{2-}]$ is derived from theoretical TA and DIC in the river-ocean endmember mixing model; the temperature- and salinity- dependent stoichiometric solubility product for aragonite, $K_{sp}^{A(*)}$, is calculated from Mucci (1983). The following equations in Mucci (1983) are involved in the determination of $K_{sp}^{A(*)}$ for each salinity: $\log K_{sp}^{A(*)} - \log K_{sp}^{A(0)} = (b_0 + b_1 SSTK + b_2/SSTK) S^{0.5} + c_0 S + d_0 S^{1.5}$ and $\log K_{sp}^{A(0)} = -171.945 - 0.077993 SSTK + 2903.293/SSTK + 71.595 \log SSTK$; SSTK is the sea surface temperature in Kelvin (i.e., SSTK = 302.15 K); b, c, and d are constants reported in Mucci (1983).

| $\Delta$TA | $\Omega_A$ (at various salinities) for Sep-2011 | | | | | |
|---|---|---|---|---|---|---|
| ($\mu$mol kg$^{-1}$) | S=0 | S=5 | S=10 | S=20 | S=30 | S=35 |
| **0** | 0.0 | 0.1 | 0.5 | 1.8 | 3.2 | 3.8 |
| **20** | 0.0 | 0.1 | 0.7 | 2.1 | 3.4 | 4.0 |
| **50** | 0.0 | 0.3 | 1.0 | 2.4 | 3.8 | 4.4 |
| **100** | 0.1 | 1.0 | 1.7 | 3.0 | 4.4 | 5.0 |
| **200** | 2.9 | 2.6 | 3.1 | 4.3 | 5.6 | 6.2 |
| **260** | 5.1 | 3.6 | 4.0 | 5.1 | 6.4 | 6.9 |

**Figures**

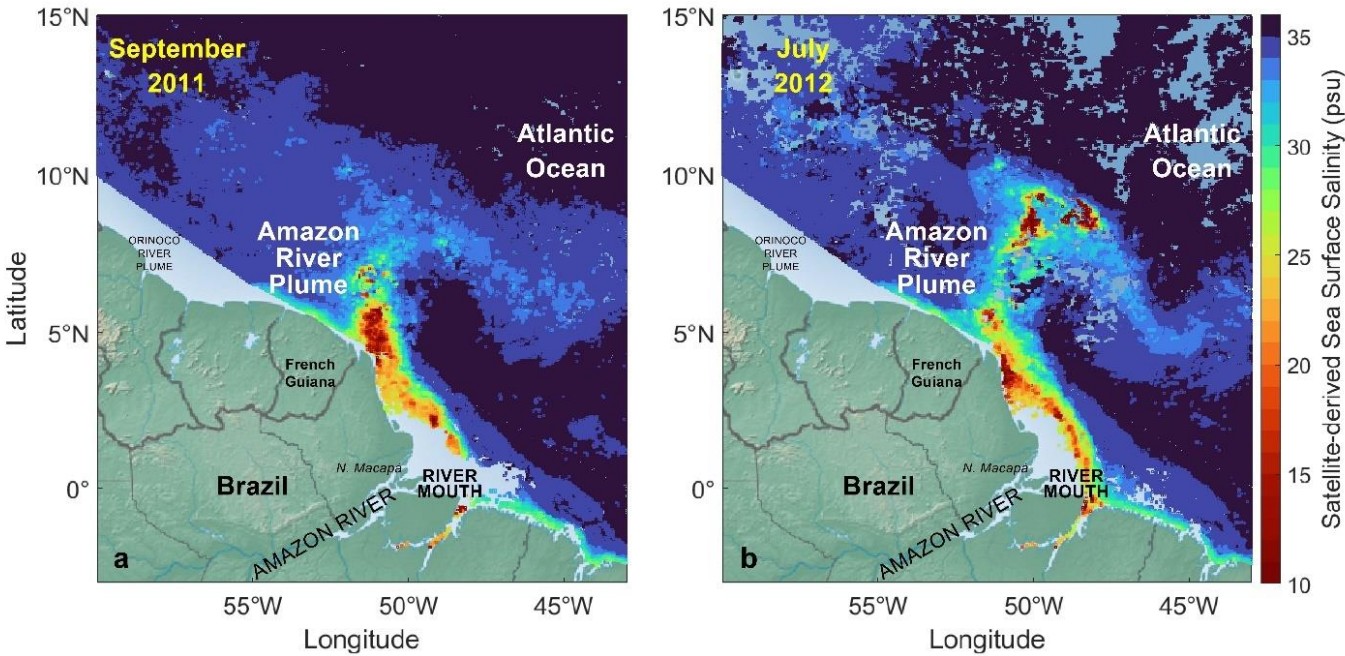

**Figure 1**. Sea surface salinity (SSS) for the Amazon River plume (15°N–3°S, 43–60°W) during (**a**) September 2011, and (**b**) July 2012, derived from remotely sensed diffuse attenuation coefficient at 490 nm. Area likely affected by the Orinoco River plume (< 150 km off the coastline between 55–60°W) is excluded from this study. Oceanic regions in light blue on the maps indicate the satellite-derived SSS data are unavailable, mostly due to muddy nearshore waters or clouds. See Mu et al. (2021) for further details.

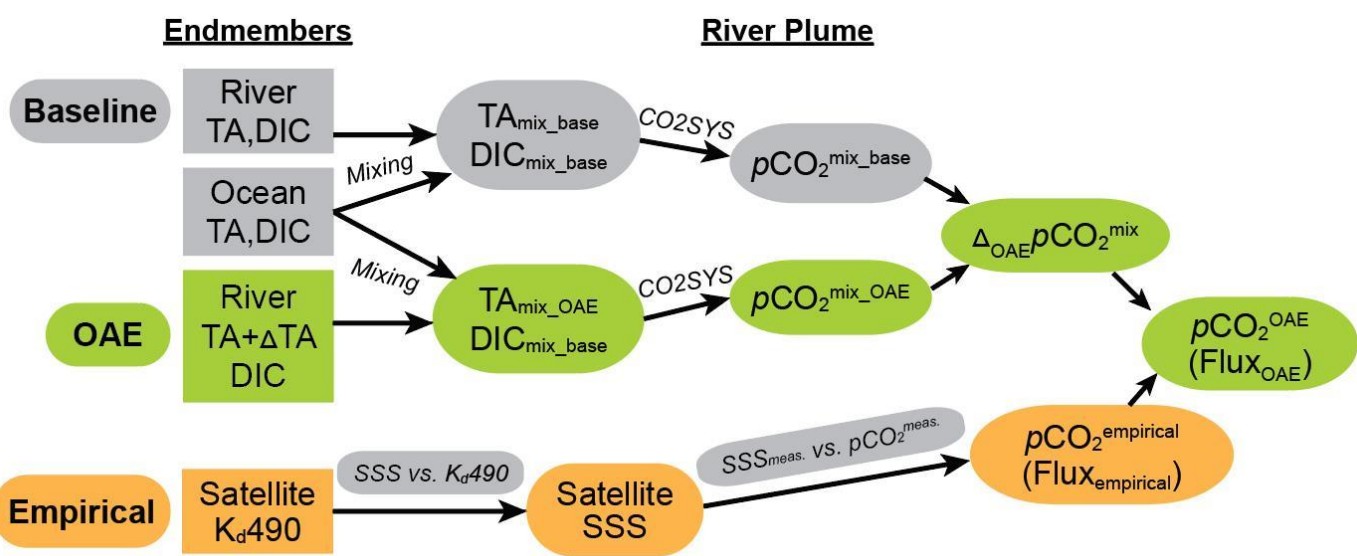

**Figure 2**. Schematic diagram for the method used to calculate carbonate chemistry and air–sea $CO_2$ fluxes before and after a hypothetical alkalinity addition (OAE) in the Amazon River-ocean continuum. The baseline and OAE pathways (gray and green) use the conservative mixing model (Equations 3-6) and CO2SYS to calculate the baseline and hypothetical OAE $pCO_2$. The difference between the baseline and OAE mixed models ($\Delta_{OAE}pCO_2^{mix} = pCO_2^{mix\_OAE} - pCO_2^{mix\_base}$) is added to the empirically-derived $pCO_2$ at every SSS. The resulting $pCO_2^{OAE}$ is used to calculate the air-sea $CO_2$ flux across the surface of the plume for each hypothetical OAE experiment.

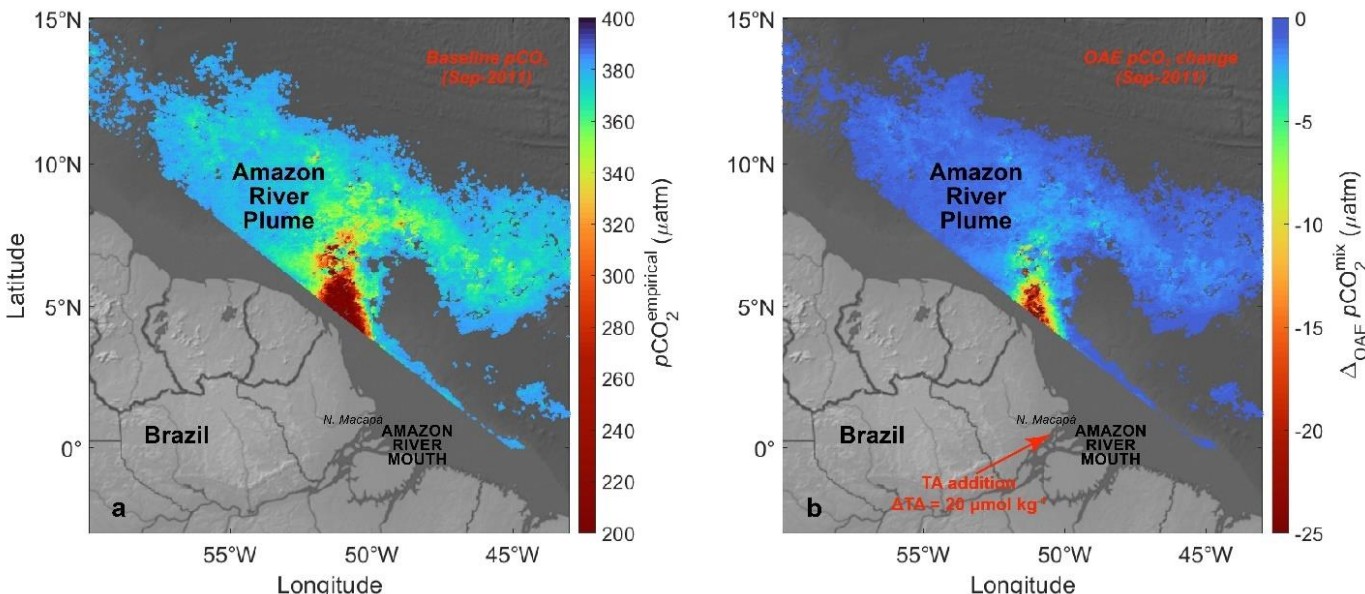

**Figure 3**. $pCO_2$ at the surface of the Amazon River plume and its predicted response to alkalinity addition based on satellite-derived SSS. (**a**) Spatial distribution of unperturbed $pCO_2^{empirical}$ ($\mu$atm) at the surface of the Amazon River plume ($15 < $ SSS $< 35$) during September 2011, with SSS outside of the plume range removed from this map as the remotely sensed $K_d490$ vs SSS regression is sufficiently robust only when $15 < $ SSS $< 35$. (See Figure 2 and Section 2.2 for details on how the mapped quantity is defined, and Mu et al. (2021) for additional details). (**b**). Predicted changes in the plume surface $pCO_2$ (i.e., $\Delta_{OAE}pCO_2^{mix}$ from Equation 9) due to addition of 20 $\mu$mol kg$^{-1}$ of alkalinity at the Amazon River mouth during September 2011.

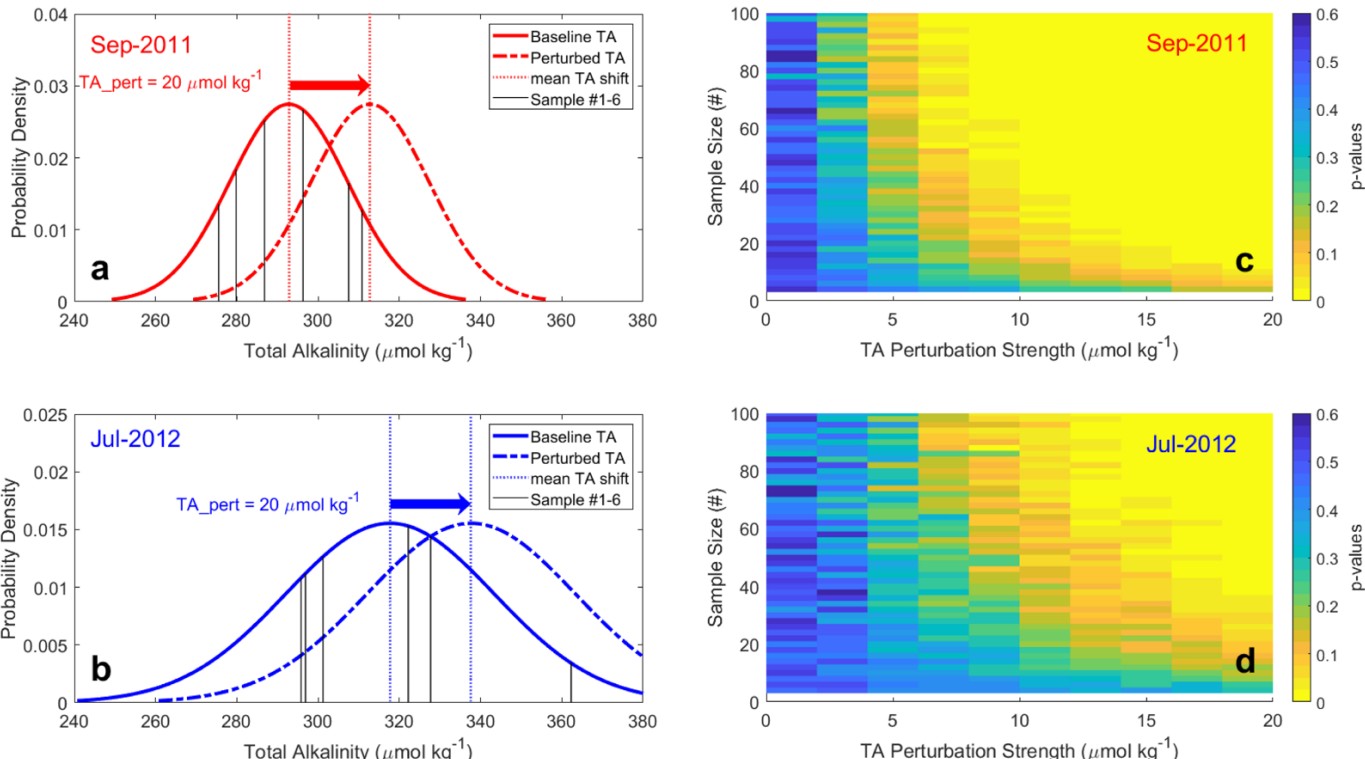

**Figure 4**. Detectability of hypothetical TA enhancement relative to background variability. (**a**)(**b**). The theoretical shift of TA ($\mu$mol kg$^{-1}$) means and distributions for September 2011 and July 2012 due to the addition of 20 $\mu$mol kg$^{-1}$ of TA at the Amazon River mouth. Black lines indicate the *in situ* TA measurements at the mouth on which the baseline data distributions are based. The standard deviations in the perturbation scenarios are assumed the same as those from the baseline. (**c**)(**d**). The p-value maps from t-tests performed between the baseline and TA perturbation scenarios at various sample sizes and TA perturbation strengths for September 2011 and July 2012. Areas in yellow ($p < 0.1$) indicate conditions where an analyst would conclude that the TA perturbation was detected relative to the baseline condition with 90% certainty.

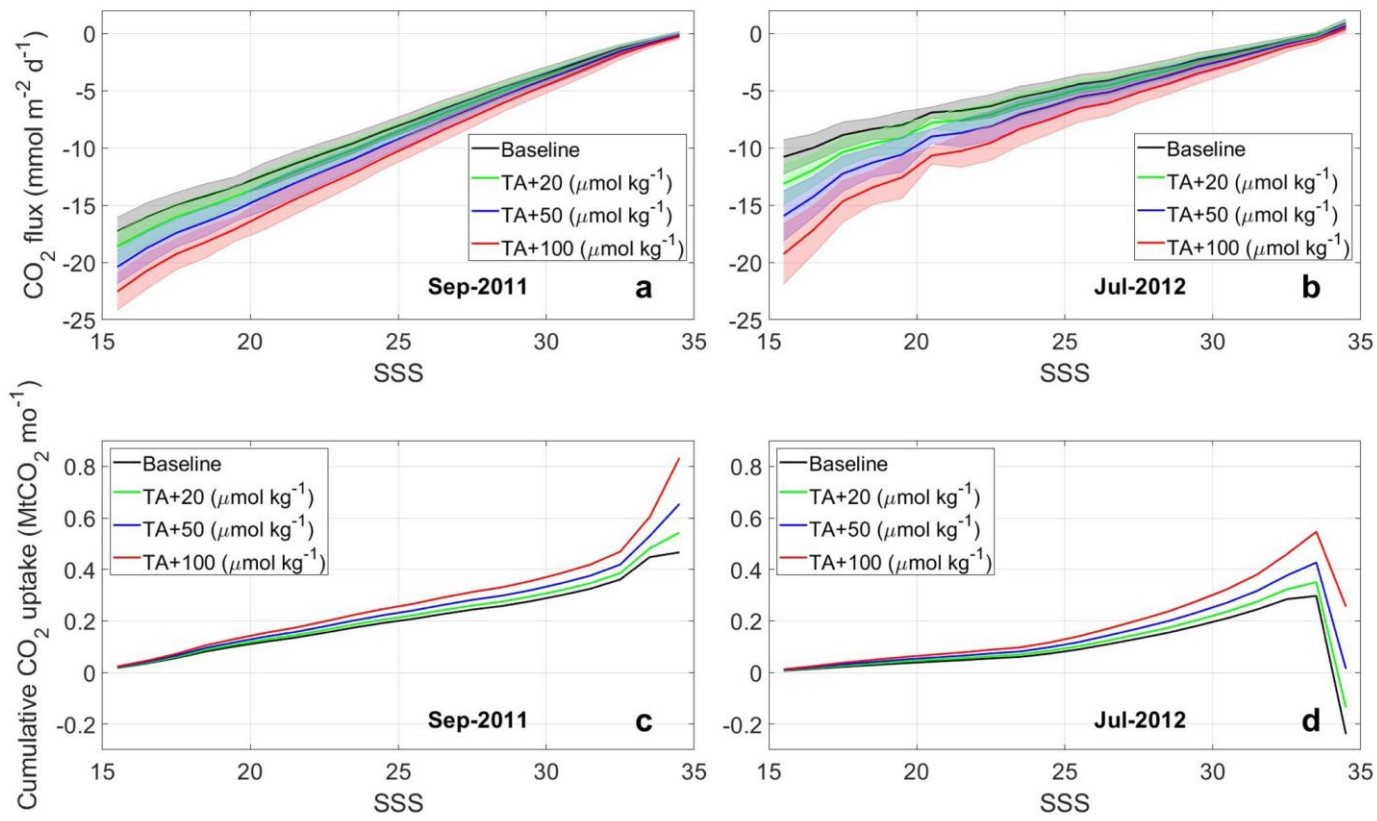

**Figure 5**. The impact of TA addition on ocean uptake of $CO_2$ across the Amazon River plume. (Top) Air-sea $CO_2$ flux density (mmol $m^{-2}$ $d^{-1}$) in the Amazon River plume at different sea surface salinities for the baseline and multiple TA perturbation scenarios in **(a)** September 2011 and **(b)** July 2012. Standard deviations of the fluxes within each salinity band (1 psu apart) are represented by the shaded areas. Negative values indicate atmospheric $CO_2$ sinks. (Bottom) Cumulative ocean $CO_2$ uptake (MtCO$_2$ mo$^{-1}$) in the Amazon plume due to TA additions in **(c)** September 2011 and **(d)** July 2012. SSS ranging between 15 and 35 is used while SSS < 15 and > 35 is omitted, because the remotely sensed K$_d$490 vs SSS regression is sufficiently robust only when 15 < SSS < 35 according to the Amazon River plume definition from Mu et al. (2021).