# Peer review of "Considerations for hypothetical carbon dioxide removal via alkalinity addition in the Amazon River watershed"

_EGUsphere, 2022_

## Author Comment (AC1)

**Response to Reviewers**

We thank both reviewers for their thoughtful and constructive feedback, which has substantially improved this manuscript. Our responses are interwoven as text in blue italics in the reviewer comments below.

**Reviewer 1.**

**1.0.** This study performs a thought experiment to explore the CO2 removal effect via ocean alkalinity enhancement (OAE) by the hypothetical quicklime addition in the Amazon River watershed. The calculation results suggest that the total carbon uptake in the Amazon River Plume is ~ 0.07-0.1 MtCO2/month. A Monte Carlo simulation is made to assess the detectability of alkalinity perturbation, which shows that the detectability depends on the perturbation strength, the sample sizes and background alkalinity variability. This paper also discusses other potential issues related with the OAE deployment, including secondary mineral precipitation, ecological consequence, and heat release during quicklime dissolution. In summary, the authors argue that the proposed thought experiment could serve as a great starting point for investigating further the feasibility of using OAE for CO2 removal.

I find this study interesting, and the argument is well organized and convincing. I do have, however, several concerns that need to be addressed first, which I summarize in the following.

**We thank the reviewer for the compliment and gladly address all the comments below.**

**1.1.** Carbonate system calculation for excess CO2 uptake estimate. The author applies a subtle method to estimate the CO2 uptake via the quicklime addition. One fundamental assumption is the constant DIC before any significant air-sea equilibrium occurs (Line 123). It is supported by that the air-sea equilibration takes weeks in the study region while CaO dissolution happens on hour-scales. Correct me if I am wrong, but I think this assumption needs more justification. The CO2 uptake estimate is based on the whole Amazon River plume region. Although CaO dissolves fast, the timescale for spreading of alkalinity perturbation along the plume may be comparable to that required for the air-sea equilibration. It is reasonable to expect a DIC increase from the river gateway to the oceanic part. With an increase of DIC, the pCO2 in the distal plume seawater will increase, which will reduce the CO2 uptake in this region. According to Figure 5, the plume at near-oceanic salinity level contributes to the majority of CO2 uptake due to the large area. Thus, I suggest the authors discuss or test the sensitivity of CO2 uptake estimate to the CO2 exchange between ocean and atmosphere along the plume route.

**We thank the reviewer for raising this point, which prompted us to clarify the assumption further in the text.**

The reviewer is correct about the implication of this assumption. We assume that the  $pCO_2$  deficit inferred from the mixing model + OAE deployment is "static" in the Figure 3 map and use the resulting static surface seawater  $pCO_2$  field to calculate the fluxes across the entire plume (Figure 5 and Table 2). In reality, exchange with the atmosphere along the path of Lagrangian parcels in the plume means that the  $pCO_2$  deficit will slowly be diminished as

the DIC increases. Thus, the true  $pCO_2$  map and the integrated flux calculated from the  $pCO_2$  deficit map will differ from our calculations.

However, we argue that this error is unlikely to strongly bias our results or change our overall conclusions, as the time elapsed between the river discharge at the mouth and mixing with ocean water to relatively high salinity (SSS > 15) is short. Comparing Figure 5 in Coles et al. (2013; copied below) to maps of SSS (e.g., Figure 1 from the manuscript), it is clear that a Lagrangian parcel discharged at the mouth of the Amazon spends little time (< approximately 2 weeks) in the fresh part of the plume (SSS < 30). The implication is that mixing of fresh river discharge with ocean water happens quickly relative to air-sea CO2 exchange. Trying to add more precision to these statements would likely require analysis of a realistic circulation-biogeochemical model, which could comprehensively capture the transport, air-sea exchange, and other chemical transformations. We leave such analysis for future work. We have added this explanation to the text.

*Figure 5* from Coles et al. (2013). Time in days since a Lagrangian particle entered the river mouth (in days). Panels on the left are made from a general circulation model; on the right shows observational surface drifters that entered the river mouth.

**1.2.** The pCO2 baseline derived from the mixing model is essential for the reliability of CO2 uptake estimate. However, the robustness of pCO2 baseline lacks discussion in the paper. I suggest plotting a figure (may be put in the extended figures) to compare the pCO2-baseline with the pCO2-empirical, which will make the performance of a simple mixing model more accessible to the reader.